# Identification of the preoperative and perioperative factors that predict postoperative endothelial cell density after Descemet membrane endothelial keratoplasty: A retrospective cohort study

**Dimitri Chaussard[1], Florian Bloch[1], Arpiné Ardzivian Elnar[2], Yinka Zevering[1], Jean-Charles Vermion[1], Rémi Moskwa[1], Jean-Marc Perone[1,3]***

**1** Ophthalmology Department, Mercy Hospital, Regional Hospital Center (CHR) of Metz-Thionville, Metz, France, **2** Clinical Research Support Unit, Mercy Hospital, Regional Hospital Center (CHR) of Metz-Thionville Regional Hospital Center, Metz, France, **3** Institut Jean Lamour, Lorraine University, Nancy, France

* jm.perone@chr-metz-thionville.fr

**Data Availability Statement:** The datasets generated during and/or analyzed during the

## Abstract

Low postoperative endothelial-cell density (ECD) plays a key role in graft failure after Descemet-membrane endothelial keratoplasty (DMEK). Identifying pre/perioperative factors that predict postoperative ECD could help improve DMEK outcomes. This retrospective study was conducted with consecutive adult patients with Fuchs-endothelial corneal dystrophy who underwent DMEK in 2015–2019 and were followed for 12 months. Patients underwent concomitant cataract surgery (triple-DMEK) or had previously undergone cataract surgery (pseudophakic-DMEK). Multivariate analyses assessed whether: patient age/sex; graft-donor age; preoperative ECD, mean keratometry, or visual acuity; triple DMEK; surgery duration; surgical difficulties; and need for rebubbling predicted 6- or 12-month ECD in the whole cohort or in subgroups with high/low ECD at 6 or 12 months. The subgroups were generated with the clinically relevant threshold of 1000 cells/mm$^2$. Surgeries were defined as difficult if any part was not standard. In total, 103 eyes (95 patients; average age, 71 years; 62% women) were included. Eighteen eyes involved difficult surgery (14 difficult graft preparation or unfolding cases and four others). Regardless of how the study group was defined, the only pre/perioperative variable that associated significantly with 6- and 12-month ECD was difficult surgery (p = 0.01, 0.02, 0.05, and 0.0009). Difficult surgery also associated with longer surgery duration (p = 0.002). Difficult-surgery subgroup analysis showed that difficult graft dissection associated with lower postoperative ECD (p = 0.03). This association may reflect endothelial cell loss due to excessive graft handling and/or an intrinsic unhealthiness of the endothelial cells in the graft that conferred unwanted physical properties onto the graft that complicated its preparation/unfolding.

current study are not publicly available according to French Law No. 2018-493 of June 20, 2018 on the protection of personal data (The General Data Protection Regulation (Regulation (EU) 2016/679) (GDPR: article 9) but are available from the Clinical Research Support Platform (Plateforme d'Appui à la Recherche Clinique [PARC]) of the Regional Central Hosital (CHR) of Metz-Thionville on reasonable request (email: projet-recherche@chr-metz-thionville.fr, tel: +33 3 87 17 98 82). All non-archived data is subject to daily backups while all archived data is subject to duplicate storage at two different sites. This data processing is compliant with a baseline reference methodology (MR-004) for which the CHR Metz-Thionville signed a compliance commitment on October 8, 2018.

**Funding:** The author(s) received no specific funding for this work.

**Competing interests:** The authors have declared that no competing interests exist.

## Introduction

In 1998–2002, endothelial keratoplasty became a feasible alternative to penetrating keratoplasty (PKP) for corneal endothelial disorders such as Fuchs endothelial corneal dystrophy (FECD) and moderate bullous keratopathy (BK). At this timepoint, Melles introduced deep lamellar endothelial keratoplasty (DLEK). This procedure involves (i) dissecting off a posterior lamellar disc from the diseased recipient cornea; (ii) inserting a folded donor disc consisting of posterior stroma, Descemet membrane, and endothelium *via* a self-healing tunnel incision; and (iii) unfolding the disc and appending it to the recipient cornea with an air bubble [1–3]. This method was then further refined by Melles and others [4–10], thus generating first Descemet stripping endothelial keratoplasty (DSEK), then Descemet stripping automated endothelial keratoplasty (DSAEK), and most recently, Descemet membrane endothelial keratoplasty (DMEK), where the donor disc consists only of Descemet membrane and endothelium [10, 11]. Multiple studies then showed that compared to PKP, endothelial keratoplasty leads to more predictable refractive outcomes, increased tectonic stability, faster postoperative visual rehabilitation, and overall better and faster visual recovery. This reflects in part the absence of graft sutures, which can induce ocular surface complications and high or irregular astigmatism [11–15].

A common concern regarding all keratoplasty techniques is that they induce a steep short-term decrease in endothelial cell density (ECD) that eventually slows to low rates [16–20]. For example, in DMEK, 30–40% of endothelial cells are lost in the first 12 months, after which there is a stable annual loss of 7% [21]. This is significant because the Cornea Donor Study and its secondary analyses showed that endothelial cell loss (ECL) predicts long-term graft failure after PKP [22–25]. This reflects the fact that loss of endothelial cells can compromise the ability of the cornea to control stromal hydration and therefore maintain corneal transparency [26]. A meta-analysis of 10 studies suggests that PKP associates with more ECL than DSAEK [27] while other meta-analyses suggest that DSAEK and DMEK are similar in terms of ECL [28–31]. This further supports the transition from PKP to endothelial keratoplasty, particularly to DMEK, since it associates with significantly better postoperative best spectacle-corrected visual acuity (BSCVA) than DSAEK. This superiority may reflect the fact that DMEK grafts lack the posterior stroma that is found in DSAEK grafts [28–31], which could generate irregularities and incongruent stromal collagen fibers in the interface between the host stroma and donor [32]. Moreover, DMEK yields fewer high order aberrations [33–35] and may even be safe for regrafting after DSAEK has yielded poor visual results [36, 37]. In addition, studies show that DMEK associates with promising medium-term (8 years) endothelial survival [38] and faster visual rehabilitation than other keratoplasty techniques [39–42].

With the aim of further improving DMEK outcomes, several studies have comprehensively sought to identify the preoperative and perioperative factors that associate with low ECD after DMEK (S1 Table) [20, 43–48]. However, there are significant variations between these studies in terms of the predictive factors that have been found: for example, older patient age associates with worse ECL in two comprehensive studies [20, 45] but not in five others [43, 44, 46–48]. To further clarify this issue, we conducted a retrospective cohort study to determine which of a large range of preoperative and perioperative factors can predict ECD at 6 and 12 postoperative months.

## Materials and methods

### Study design and ethics

This retrospective single-center cohort study was performed in the Ophthalmology Department of Metz-Thionville Regional Hospital Center, Grand Est, France and was approved by

the Ethics Committee of the French Society of Ophthalmology (IRB 00008855). All procedures conformed to the principles of the Declaration of Helsinki. All patients were informed before surgery that their surgery-related data might be used for research purposes. All consented to this possibility. The consent procedure was conducted in accordance with the reference methodology MR-004 of the National Commission for Information Technology and Liberties of France (No. 588909 v1).

## Patient selection

The study cohort consisted of all consecutive adult ($\geq$18 years) patients with FECD who underwent DMEK between October 2015 and October 2019 and who were followed up for at least 12 months. Patients either underwent concomitant cataract surgery (defined as triple-DMEK) or had previously undergone cataract surgery (defined as pseudophakic-DMEK). The surgery was usually deemed necessary when the BSCVA reached 0.3 logMAR. Exclusion criteria were prior eye surgery other than cataract surgery, indications other than FECD, the operative eye had important retinal or optic nerve diseases or amblyopia, and intraoperative complications.

## Preoperative and postoperative testing

Before and 6 and 12 months after surgery, all patients underwent a macular OCT Scan (NIDEK, Tokyo, Japan) to detect retinal anomalies, specular microscopy (NIDEK CEM-530; NIDEK, Tokyo, Japan) to measure ECD, and measurement of BSCVA (in Monoyer scale, followed by conversion into logMAR).

## Graft preparation, surgical techniques, postoperative treatment, and follow-up

When surgery was planned with the patient in consultation, inferior iridotomy with a Nd: YAG Laser (Laser ex-Super Q; Ellex Europe, Medical Quantel, Cournon-d'Auvergne, France) was conducted to prevent pupillary block during and after DMEK surgery.

All surgeries were performed by an experienced surgeon (JMP). General anesthesia was used for all but two patients, who had contraindications and therefore underwent surgery under a peribulbar block (a 50:50 mixture of ropivacaine 7.5 mg/mL and lidocaine 2% delivered with a 23-gauge Atkinson canula).

All grafts were issued by either of two French regional tissue banks (Nancy or Besançon). They were stored in organ culture medium (Eurobio) at 31˚C until transplantation and had the requested ECD ($>$2400 cells/mm$^2$), which was measured by the eye bank. To prepare the grafts for DMEK, they were first trephined with a Hanna trephine (Busin Punch 17200D 8mm single use; Moria SA, Antony, France) and the Descemet membrane-endothelium complex was dissected off with a disposable curved nontoothed forceps (Single Use Tying Forceps Curved 5mm Platform 17501; Moria SA, Antony, France) under a microscope. The resulting roll was stained with Trypan Blue (VisionBlue, 0.5-ml syringe; D.O.R.C. Dutch Ophthalmic Research Center, Zuidland, Netherlands), marked with a "F" on its stromal side, and inserted into a D.O.R.C. injectable system (30G Curved cannula for air injection; D.O.R.C. Dutch Ophthalmic Research Center, Zuidland, Netherlands).

Paracentesis was performed using a 2.2-mm blade (Securityblade BD, Xstar 2.2-mm, 45 degrees, 37822; Beaver-Visitec International, Waltham, USA) and a Worst 15 blade (ophthalmic knife 15 degrees; ALCON, Ruel Malmaison, France). A 9-mm central descemetorhexis was then performed under sterile air with an inverted Sinskey Price hook (Single Use Price Reverse Hook Sim 17302; Moria SA, Antony, France) and an inverted spatula (90$^{th}$ single use

Spatula 17303; Moria SA, Antony, France). The main incision was enlarged to 4 mm and the graft was inserted into the anterior chamber *via* the D.O.R.C injectable system. Two 27-gauge Rycroft cannulas were manipulated to center the graft and orient it such that the rolls of the scroll faced upward. To keep the graft in place, a sterile air bubble was injected underneath the graft. If the epithelium seemed even slightly pathological, it was removed. A 10–0 Nylon suture was placed at the main incision.

If cataract surgery was indicated, it was conducted before DMEK using the standard supra-capsular "garde à vous" technique [49] with implantation of a Zeiss CT Asphina 409MV intra-ocular lens. The refractive target in triple DMEK was a residual myopia of about -0.50 to -1.00 D to compensate for the expected hyperopic shift induced by DMEK surgery [50, 51].

The postoperative regimen consisted of drops composed of 0.1% dexamethasone, neomy-cin, and polymyxin B (Maxidrol; ALCON, Rueil Malmaison, France) four times daily for 4 weeks. It was then gradually tapered and followed with a 0.1% fluorometholone-tapering (Flu-con; ALCON, Rueil Malmaison, France) regimen for at least 12 months. Patients who under-went triple DMEK also received non-steroidal anti-inflammatory drops (indomethacin) (Indocollyre; Chauvin, Montpellier, France) four times daily for 4 weeks.

The patients were hospitalized for 3 days in the Ophthalmology Unit and monitored by vis-its 1, 7, and 15 days and 1, 3, 6, and 12 months after surgery.

Patients who developed cystoid macular edema and loss of visual acuity were treated with oral acetazolamide (Diamox®; Sanofi, Gentilly, France; one 250 mg tablet 3 times/day for one month) and NSAID (indomethacin 0.1%; Chauvin, Montpellier, France; 4 times/day for one month).

Patients with allograft rejection (defined as a line of retro-descemetic precipitates) were treated for one month with combined treatment composed of an anti-inflammatory cortico-steroid ointment (Sterdex®, which contains dexamethasone with oxytetracycline; Thea, France; Clermont-Ferrand, 2 times/day) and an anti-inflammatory corticosteroid eyedrop (Maxidrol; ALCON, Rueil Malmaison, France; 12 times/day for one week then 8, 6, and 4 times/day for the second, third, and fourth week, respectively).

Rebubbling was performed immediately if anterior segment OCT (NIDEK; Tokyo Japan) showed that more than a third of the graft had detached or the graft detachment was threaten-ing the visual axis [52]. Rebubbling was conducted under topical anesthesia (0.4% oxybupro-caïne hydrochloride; Thea, Clermont-Ferrand, France) using sterile air and an operating microscope. If a fourth rebubbling session was needed, the graft was considered to have failed and the patient was scheduled for repeat keratoplasty.

## Preoperative, perioperative, and postoperative variables

The following data were obtained from the prospectively maintained clinical database: patient age and sex, donor age, graft ECD before and 6 and 12 months after surgery, BSCVA before and 6 and 12 months after surgery, preoperative mean anterior keratometry, triple DMEK, dif-ficult surgery, surgical time, and number of rebubbling sessions. DMEK surgery was labelled as "difficult" if any part of it was not standard. Examples include difficult graft dissection, graft defects, abnormal graft elasticity, difficulties unfolding the scroll in the anterior chamber, or any condition hindering the normal course of surgery or requiring more maneuvers than usual. Surgical time was defined as the time taken from starting graft preparation to placing the suture.

## Statistical analysis

All data were expressed as mean and standard deviation (range), as medians (IQR: 10; 25; 75; 90), or *n* (%). The relationships between 6- and 12-month ECD in the whole cohort and

patient, graft, and operative variables (including difficult surgery) were determined with simple and multiple linear regression models. All 10 variables examined in the univariate analyses were considered in the multivariate analyses because the sample size was sufficient for this number of factors. The 6- and 12-month cohorts were also each divided into two subgroups depending on whether their postoperative ECD was <1000 or ≥1000 cells/mm$^2$. This threshold was selected because several ophthalmology societies consider it to be a risk threshold of corneal decompensation [53, 54]. The high/low postoperative ECD subgroups were then compared in terms of preoperative/perioperative variables by Mann-Whitney $U$ test (quantitative variables) or Fisher's exact test (qualitative variables) and then with logistic regression. Subgroup analysis was also conducted to determine whether eyes with different types of surgical difficulties differed from eyes with non-difficult surgery in terms of postoperative ECD: here, Kruskal-Wallis test followed by Mann-Whitney $U$ test was used. Student's t-test was used to evaluate whether rebubbling associated with mean anterior keratometry. All statistical analyses were conducted using the Statview 4.5 statistical package (Abacus Concepts, Inc., New Jersey, USA). P values <0.05 were considered significant.

## Results

In total, 141 eyes underwent primary or triple DMEK during the study period. Of these, 38 were excluded because the indication was not FECD ($n$ = 19) or there was a history of an ocular intervention other than cataract surgery ($n$ = 11), another corneal, retinal, or optic nerve disease ($n$ = 4), or intraoperative complications ($n$ = 4; two extreme brunescent cataracts that led to high effective phaco time, and two eyes that were converted to DSAEK due to extensive progression of stromal edema between the last visit and the surgery). Thus, 103 eyes of 95 patients were included in the study (Fig 1).

### Patient and eye characteristics

Average patient age was 71 years and 62% were women. Preoperative mean anterior keratometry was 44 D. Average graft-donor age was 76 years. Triple DMEK was conducted in 40 eyes and pseudophakic DMEK in 63. Average surgery duration was 35 min and 18 (17%) surgeries were deemed difficult. The average durations of the difficult and not difficult surgeries were 40.3±10.1 and 33.2±8.2 min, respectively (p = 0.002). The causes of difficult surgery were difficult graft dissection ($n$ = 7), difficult unfolding of the graft in the anterior chamber ($n$ = 7), anterior segment hazards (iridocorneal synechiae and two cases that required premature postoperative YAG treatment) ($n$ = 2), air bubble under the iris during surgery ($n$ = 1), and a patient on peribulbar block moving during surgery ($n$ = 1). Average graft ECDs before and 6 and 12 months after surgery were 2557, 1403 (45% ECL), and 1244 (51% ECL) cells/mm$^2$, respectively. Preoperative BSCVA was 0.64 logMAR on average, and surgery yielded excellent visual outcomes: average BSCVAs at 6 and 12 months were 0.09 and 0.05 logMAR, respectively. Rebubbling was conducted in 34% of cases. Most only required a single rebubbling session (25% of all eyes). More than one rebubbling session was needed in 9% of all eyes. None of the eyes required more than three rebubbling sessions (Table 1). Cystoid macular edema and graft rejection were observed in three and one patients, respectively. In total, 10% of the grafts failed at 12 months and the patients underwent repeat keratoplasty.

### Relationships between 6- and 12-month endothelial cell density in the whole cohort and preoperative and perioperative variables

Simple and multiple linear regression analyses with the whole cohort showed that 6-month ECD associated strongly with difficult surgery (p = 0.005 and 0.01, respectively) (Table 2). This

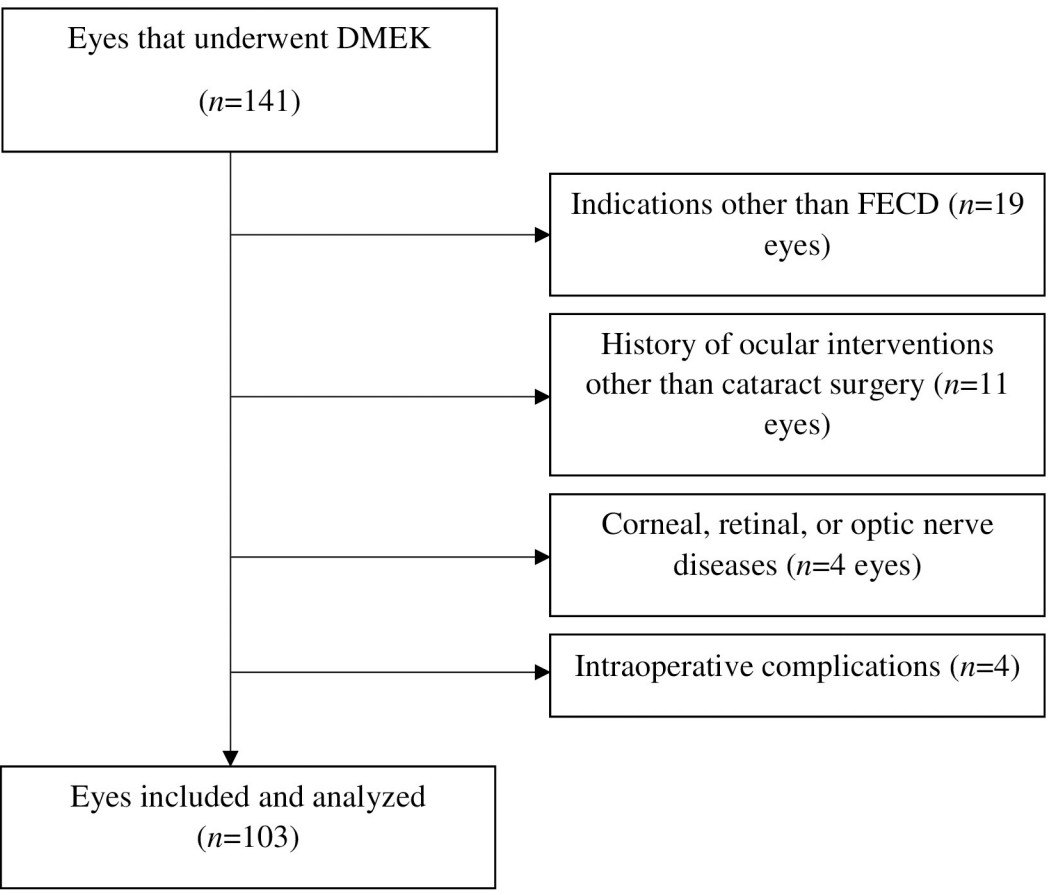

**Fig 1. Flow chart showing patient disposition in the study.**

association was also present at 12 months (p = 0.014 and 0.02, respectively) (Table 3). We then categorized the difficult surgeries as graft dissection difficulties ($n = 7$), graft unfolding difficulties ($n = 7$), and other difficulties ($n = 4$), and compared them to the 85 DMEK eyes that did not involve surgical difficulties. Kruskal-Wallis test showed that the four groups differed significantly in terms of 12-month ECD (p = 0.03). Mann-Whitney $U$ tests then showed that graft dissection difficulties associated with significantly lower median 12-month ECD than non-difficult DMEK (945 $vs.$ 1331 cells/mm$^2$; p = 0.01). Graft unfolding difficulties also tended to associate with lower 12-month ECD (800 $vs.$ 1331 cells/mm$^2$; p = 0.07). The other complications (anterior segment hazards, air bubble below iris, and patient movements) did not affect 12-month ECD (1500 $vs.$ 1331 cells/mm$^2$) (Fig 2). Kruskal-Wallis test also showed that the non-difficult and three difficult subgroups tended to differ in median 6-month ECD (1470 $vs.$ 1009 $vs.$ 1000 $vs.$ 1025 cells/mm$^2$) but these differences did not achieve statistical significance (p = 0.06) (Fig 2).

In relation to the other variables, preoperative ECD tended to associate with postoperative ECD at 6 months on univariate analysis (p = 0.07) but not on multivariate analysis (p = 0.25) and not at 12 months (p = 0.20 and 0.50, respectively). Other variables did not associate with postoperative ECD, including rebubbling (Tables 2 and 3).

**Table 1. Demographic and clinical characteristics of the patients and eyes undergoing Descemet membrane endothelial keratoplasty.**

| Variable | $n$ (%) or mean ± standard deviation |
|---|---|
| No. of eyes (patients) | 103 (95 patients) |
| Patient sex, F/M | 64/39 |
| Age, years | |
| Patient age at surgery | 70.5 ± 9.0 (48.0–90.0) |
| Graft donor age | 75.5 ± 11.0 (30.0–99.0) |
| Mean anterior keratometry, D | 43.93 ± 1.54 (39.87–48.61) |
| ECD, cells/mm$^2$ | |
| Preoperative (graft) | 2557 ± 200 (2000–2950) |
| 6 postoperative months | 1403 ± 429 (535–2500) |
| 12 postoperative months | 1244 ± 416 (509–2400) |
| BSCVA, logMAR | |
| Preoperative | 0.64 ± 0.29 (0.30–2.00) |
| 6 postoperative months | 0.09 ± 0.12 (-0.10–0.50) |
| 12 postoperative months | 0.05 ± 0.10 (-0.20–0.40) |
| Surgery and postoperative complications | |
| Triple procedure | 40 (39%) |
| Difficult surgery | 18 (17%) |
| Surgery time, min | 34.5 ± 8.9 (20.0–60.0) |
| Rebubbling | |
| None | 68 (66%) |
| One | 26 (25%) |
| More than one | 9 (9%) |

BSCVA, best spectacle-corrected visual acuity; ECD, endothelial corneal density; F, female; M, male.

## Comparison of patients with low and high postoperative ECDs in terms of preoperative and perioperative variables

We then divided both the 6- and 12-month cohorts into two subgroups by applying a postoperative ECD cut-off of 1000 cells/mm$^2$. This value was selected because several ophthalmology societies consider it to be a risk threshold of corneal decompensation [53, 54]. Compared to patients with a high 6-month ECD ($\geq$1000 cells/mm$^2$), patients with a low 6-month ECD ($<$1000 cells/mm$^2$) tended to associate with a higher surgical difficulty rate (32% *vs*. 14%, p = 0.06 and 0.05 on univariate and multivariate analysis, respectively) (Table 4). This trend became significant at 12 postoperative months (37% *vs*. 8%, p = 0.001 and 0.0009, respectively) (Table 5).

Patients with a lower 6-month postoperative ECD also had significantly fewer endothelial cells in the graft before surgery on univariate analysis (2487 *vs*. 2576 cells/mm$^2$, p = 0.03) but this was not observed on multivariate analysis (p = 0.15) (Table 4) or at 12 postoperative months (2528 *vs*. 2569 cells/mm$^2$, p = 0.25 and 0.98 on univariate and multivariate analysis, respectively) (Table 5). Patients with a lower 12-month postoperative ECD also underwent significantly longer surgery on univariate analysis (36.5 *vs*. 33.6 min, p = 0.03) but this was not observed on multivariate analysis (p = 0.53) (Table 5) or at 6 postoperative months (36.1 *vs*. 34.0 min, p = 0.11 and 0.51 on univariate and multivariate analysis, respectively) (Table 4).

Other variables did not associate with postoperative ECD in this analysis, including rebubbling (Tables 4 and 5).

**Table 2. Simple and multiple linear regression analysis of the relationship between endothelial cell density at 6 postoperative months and preoperative patient or graft or operative variables.**

| Variable | Simple regression | | | | | 95% CI | | Multiple regression | | | | 95% CI | |
|---|---|---|---|---|---|---|---|---|---|---|---|---|---|
| | B | SE | t | *p*-value | $R^2$ | Lower | Upper | B | SE | t | *p*-value | Lower | Upper |
| Patient age, years | 1.93 | 4.77 | 0.40 | 0.69 | 0.002 | -7.53 | 11.39 | -0.38 | 5.06 | -0.10 | 0.94 | -10.48 | 9.52 |
| Patient sex | 16.62 | 87.60 | 0.19 | 0.85 | 365.4E-6 | -157.15 | 190.39 | 22.83 | 86.99 | 0.30 | 0.79 | -145.60 | 197.46 |
| Donor age, years | 3.68 | 3.85 | 0.95 | 0.34 | 0.009 | -3.97 | 11.32 | 5.36 | 3.86 | 1.42 | 0.17 | -2.19 | 13.07 |
| Mean anterior keratometry, D | 14.53 | 27.64 | 0.53 | 0.60 | 0.003 | -40.30 | 69.36 | 11.87 | 28.63 | 0.42 | 0.68 | -44.50 | 68.65 |
| Preoperative ECD, cells/mm$^2$ | 0.39 | 0.21 | 1.86 | 0.07 | 0.033 | -0.03 | 0.81 | 0.27 | 0.23 | 1.18 | 0.25 | -0.18 | 0.71 |
| Preoperative BSCVA, logMAR | 168.72 | 148.10 | 1.14 | 0.26 | 0.013 | -125.06 | 462.51 | 77.24 | 161.34 | 0.51 | 0.63 | -235.30 | 398.49 |
| Triple procedure | -111.97 | 86.48 | -1.30 | 0.20 | 0.007 | -283.52 | 59.59 | -105.38 | 100.81 | -1.10 | 0.30 | -308.00 | 89.06 |
| Difficult surgery | -309.31 | 107.59 | -2.88 | **0.005** | 0.076 | -522.73 | -95.89 | -317.90 | 121.47 | -2.66 | **0.010** | -557.36 | -80.66 |
| Surgery time, min | -4.80 | 4.77 | -1.01 | 0.32 | 0.010 | -14.26 | 4.66 | 3.72 | 6.02 | 0.66 | 0.54 | -7.79 | 15.58 |
| Any rebubbling | -66.51 | 65.28 | -1.02 | 0.31 | 0.010 | -196.01 | 62.99 | -69.15 | 67.94 | -1.02 | 0.31 | -204.10 | 65.79 |

All data are presented as slope unstandardized coefficient (B) and standard error (SE) of the regression. Statistically significant relationships between ECD and preoperative/perioperative variables (p<0.05) are bolded. The $R^2$ for the multiple regression model was 0.14, p = 0.15 (ANOVA).
BSCVA, best spectacle-corrected visual acuity; ECD, endothelial cell density.

## Relationship between mean anterior keratometry and rebubbling

In a side analysis, we observed that when we divided the patients according to whether they underwent no rebubbling or any rebubbling, rebubbling associated with significantly smaller mean anterior keratometry values (*i.e.* flatter cornea) compared to no rebubbling (43.44±1.41 *vs*. 44.19±1.55 D; p = 0.02).

## Discussion

The present cohort study searched for pre/perioperative factors that shape the postoperative ECD after DMEK. We observed with two statistical methods that difficult surgery associated

**Table 3. Simple and multiple linear regression analysis of the relationship between endothelial cell density at 12 postoperative months and preoperative patient or graft or operative variables.**

| Variable | Simple regression | | | | | 95% CI | | Multiple regression | | | | 95% CI | |
|---|---|---|---|---|---|---|---|---|---|---|---|---|---|
| | B | SE | t | *p*-value | $R^2$ | Lower | Upper | B | SE | t | *p*-value | Lower | Upper |
| Patient age, years | 0.91 | 4.78 | 0.19 | 0.85 | 3.60E-04 | -8.57 | 10.39 | -1.09 | 4.97 | -0.02 | 0.83 | -10.95 | 8.78 |
| Patient sex | 27.36 | 85.63 | 0.32 | 0.75 | 0.001 | -142.52 | 197.24 | 35.53 | 83.43 | 0.04 | 0.67 | -130.18 | 201.24 |
| Donor age, years | 5.63 | 3.72 | 1.51 | 0.13 | 0.022 | -1.76 | 13.01 | 7.05 | 3.69 | 0.19 | 0.06 | -0.28 | 14.37 |
| Mean anterior keratometry, D | 35.42 | 26.74 | 1.33 | 0.19 | 0.017 | -17.63 | 88.48 | 25.96 | 27.38 | 0.10 | 0.35 | -28.43 | 80.35 |
| Preoperative ECD, cells/mm$^2$ | 0.27 | 0.21 | 1.30 | 0.20 | 0.016 | -0.14 | 0.68 | 0.15 | 0.22 | 0.07 | 0.49 | -0.28 | 0.58 |
| Preoperative BSCVA, logMAR | 141.75 | 144.39 | 0.98 | 0.33 | 0.01 | -144.72 | 428.21 | 93.16 | 153.04 | 0.06 | 0.54 | -210.84 | 397.17 |
| Triple procedure | -151.89 | 83.47 | -1.82 | 0.07 | 0.032 | -317.49 | 13.70 | -128.83 | 96.04 | -0.15 | 0.18 | -319.59 | 61.93 |
| Difficult surgery | -270.34 | 107.80 | -2.51 | **0.014** | 0.059 | -484.22 | -56.46 | -279.74 | 118.13 | -0.25 | **0.02** | -514.39 | -45.09 |
| Surgery time, min | -7.46 | 4.60 | -1.62 | 0.11 | 0.026 | -16.58 | 1.66 | 1.14 | 5.69 | 0.03 | 0.84 | -10.17 | 12.45 |
| Any rebubbling | -97.30 | 63.14 | -1.54 | 0.13 | 0.023 | -222.57 | 27.98 | -79.61 | 65.79 | -0.13 | 0.23 | -210.29 | 51.07 |

All data are presented as slope unstandadized coefficient (B) and standard error (SE) of the regression. Statistically significant relationships between ECD and preoperative/perioperative variables (p<0.05) are bolded. The $R^2$ for the multiple regression model was 0.16; p = 0.08 (ANOVA).
BSCVA, best spectacle-corrected visual acuity; ECD, endothelial cell density.

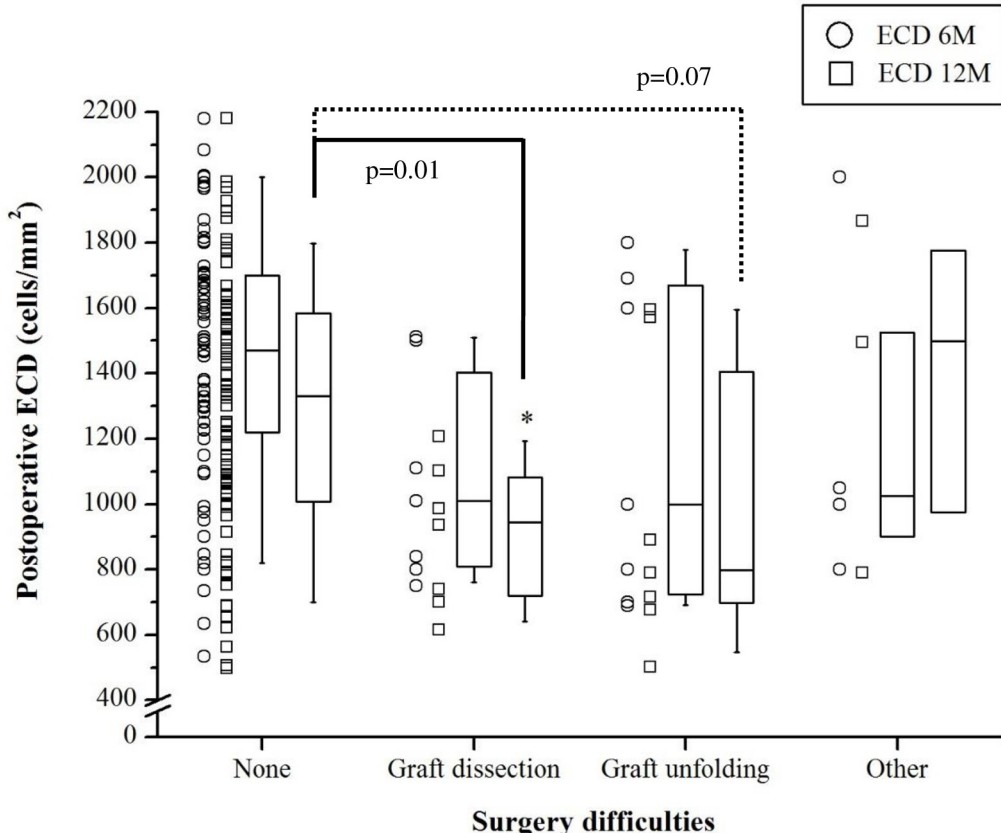

**Fig 2. Comparison of surgically difficult and uncomplicated DMEK cases in terms of ECD at 6 and 12 months.** The surgically difficult cases were divided according to the type of difficulty (difficult graft preparation, unfolding, or other). The data are presented as box-whisker plots showing the median, first and third quartiles, and maximum and minimum values. Kruskal-Wallis analyses showed that the four groups differed significantly in terms of 12-month ECD (p = 0.03). Mann-Whitney U-tests showed that 12-month ECD was significantly lower in the difficult graft preparation cases (p = 0.01) and tended to be lower in the difficult graft unfolding cases (p = 0.07) compared to the group with no surgery difficulties.

significantly with lower postoperative ECD, particularly at 12 months. DMEK surgery was deemed difficult if there was any condition that hindered or prolonged the normal surgical course. Univariate analyses also showed that low preoperative graft ECD and longer surgery associated with low preoperative ECD at 6 and 12 months, respectively. However, these relationships were not observed at the other postoperative timepoint or on multivariate analysis. All other factors, namely, patient age and sex, donor age, preoperative mean anterior keratometry, preoperative BSCVA, triple procedure, and rebubbling, did not associate with postoperative ECD.

To date, seven retrospective cohort studies have, like us, recently (2016–2021) used multivariate analysis to comprehensively assess the ability of a range of pre/perioperative variables to predict postoperative ECD/ECL [20, 43–48] (S1 Table). The findings of these studies, and other studies that are less comprehensive and/or focused instead on the relationship between pre/perioperative variables and rebubbling/graft detachment, are as follows. It should be noted that the results of these studies often agree poorly: some find certain relationships while others do not. This may reflect, at least in part, differences between the study populations in terms of surgical practice, variable measurement, postoperative timepoint, and study/statistical design.

**Table 4. Comparison of patients with 6-month ECD <1000 and ≥1000 cells/mm² in terms of preoperative and perioperative variables.**

| | 6-month ECD <1000 cells/mm² | 6-month ECD ≥1000 cells/mm² | Univariate* | Multivariate** | | |
|---|---|---|---|---|---|---|
| **Variables** | **n = 22** | **n = 81** | **p-value** | **OR (95% IC)** | **p-value** | **R** |
| Age, years | 71.1 ± 0.5 | 70.4 ± 9.1 | 0.83 | 0.97 (0.91–1.04) | 0.39 | 0.000 |
| Female sex | 16 (73%) | 48 (59%) | 0,32 | 0.54 (0.17–1.66) | 0.28 | 0.000 |
| Donor age, years | 73.9 ± 9.3 | 76.0 ± 11.5 | 0.18 | 1.03 (0.98–1.08) | 0.24 | 0.000 |
| Mean ant. keratometry, D | 43.85 ± 1.53 | 43.96 ± 1.55 | 0.91 | 1.00 (1.00–1.01) | 0.85 | 0.000 |
| Preop. ECD, cells/mm² | 2487 ± 139 | 2576 ± 210 | **0.03** | 1.43 (0.17–12.42) | 0.17 | 0.000 |
| Preop. BCVA, logMAR | 0.61 ± 0.18 | 0.65 ± 0.31 | 0.98 | 0.96 (0.65–1.43) | 0.74 | 0.000 |
| Triple procedure | 12 (55%) | 28 (35%) | 0.14 | 0.55 (0.25–1.21) | 0.16 | -0.004 |
| Difficult surgery | 7 (32%) | 11 (14%) | 0.06 | 0.41 (0.12–1.41) | 0.05 | -0.128 |
| Surgery time, min | 36.1 ± 7.2 | 34.0 ± 9.3 | 0.11 | 0.26 (0.07–1.02) | 0.51 | 0.000 |
| Rebubbling | | | 0.35 | 1.03 (0.95–1.10) | 0.14 | -0.043 |
| None | 12 (55%) | 56 (69% | | | | |
| One | 7 (32%) | 19 (23%) | | | | |
| More than one | 3 (14%) | 6 (7%) | | | | |

The data are presented as mean ± standard deviation or *n* (%). For logistic regression, data are presented as odds ratios (OR), 95% confidence interval (CI), and *p*-value.

*The two postoperative ECD subgroups were compared in terms of preoperative/operative variables by using Mann-Whitney U test (quantitative variables) or Fisher's exact test (qualitative variables).

**Logistic regression model. The $R^2$ for the multiple regression model was 0.13.

Statistically significant differences between the subgroups are bolded.

Ant., anterior; BSCVA, best spectacle-corrected visual acuity; ECD, endothelial corneal density; preop., preoperative.

**Table 5. Comparison of patients with 12-month ECD <1000 and ≥1000 Cells/mm² in terms of preoperative and perioperative variables.**

| | 12-month ECD <1000 cells/mm² | 12-month ECD ≥1000 cells/mm² | Univariate* | Multivariate** | | |
|---|---|---|---|---|---|---|
| **Variables** | **n = 30** | **n = 72** | **p-value** | **OR (95% IC)** | **p-value** | **R** |
| Age, years | 71.6 ± 7.8 | 70.1 ± 9.4 | 0.49 | 0.96 (0.90–1.02) | 0.19 | 0.000 |
| Female sex | 20 (67%) | 44 (61%) | 0.66 | 0.82 (0.29–2.29) | 0.70 | 0.000 |
| Donor age, years | 74.5 ± 9.2 | 76.0 ± 11.7 | 0.22 | 1.02 (0.98–1.07) | 0.32 | 0.000 |
| Mean ant. keratometry, D | 43.68 ± 1.47 | 44.04 ± 1.57 | 0.49 | 1.08 (0.75–1.55) | 0.69 | 0.000 |
| Preop. ECD, cells/mm² | 2528 ± 158 | 2569 ± 214 | 0.25 | 1.00 (1.00–1.00) | 0.98 | 0.000 |
| Preop. BSCVA, logMAR | 0.60 ± 1.18 | 0.66 ± 0.32 | 0.76 | 2.14 (0.28–16.30) | 0.46 | 0.000 |
| Triple procedure | 16 (53%) | 24 (33%) | 0.08 | 0.35 (0.11–1.14) | 0.08 | -0.093 |
| Difficult surgery | 11 (37%) | 6 (8%) | **0.001** | 0.09 (0.02–0.38) | **0.0009** | -0.271 |
| Surgery time, min | 36.5 ± 7.09 | 33.6 ± 9.47 | **0.03** | 1.02 (0.95–1.10) | 0.53 | 0.000 |
| Rebubbling | | | 0.16 | 0.50 (0.23–1.10) | 0.08 | -0.089 |
| None | 16 (53%) | 52 (72%) | | | | |
| One | 10 (33%) | 15 (21%) | | | | |
| More than one | 4 (13%) | 5 (7%) | | | | |

The data are presented as mean ± standard deviation or *n* (%).

*The two postoperative ECD subgroups were compared in terms of preoperative/operative variables by using Mann-Whitney U test (quantitative variables) or Fisher's exact test (qualitative variables).

**Logistic regression model. The $R^2$ for the multiple regression model was 0.18.

Statistically significant differences between the subgroups are bolded.

Ant., anterior; BSCVA, best spectacle-corrected visual acuity; ECD, endothelial corneal density, Preop., preoperative.

It is also possible that significantly-associated pre/perioperative variables relate only indirectly to the fundamental endothelial cell variable(s) that ultimately shape(s) the postoperative density and function of these cells [55, 56].

## Association between surgical difficulty and ECL

In our study, surgical difficulty associated consistently with lower ECD at two postoperative timepoints. The main causes of surgical difficulty were problematic graft dissection and unfolding, which respectively associated significantly (p = 0.01) and tended to associate (p = 0.07) with lower ECD at 12 months. We speculate that surgical difficulty reduced postoperative ECD after DMEK either because the greater graft handling directly damaged the endothelial cells and/or because the intrinsic unhealthiness of the endothelial cells in the graft conferred unwanted physical properties onto the graft that complicated its preparation and/or placement against the recipient cornea. This notion is supported by several other studies. First, two of the seven comprehensive studies examined the influence of difficult surgery on ECL [44, 45]. One, which was on a cohort of 351 eyes, found that intraoperative complications such as unfolding difficulties associated significantly with higher ECL at 48 months (Estimate 6.50, 95% confidence intervals 0.59–12.41, p = 0.032) [45]. However, the second comprehensive study, which was on 2485 eyes, did not find that intraoperative complications predicted ECL [44]. Second, Schrittenlocher et al. found that ECL depended on the learning curve, being higher in the early period [57]. Moreover, a retrospective study on 169 eyes showed that severe unfolding difficulties associated with significantly higher ECL at 3, 6, and 12 months compared to eyes with no or milder unfolding problems [58]. In addition, a retrospective study on 28 eyes observed a positive correlation between ECL and DMEK graft unfolding time [59]. A similar study on 93 eyes did not observe this association but this may reflect the fact that all surgeries were uncomplicated [60].

Third, others have noted that surgical difficulties, including poor graft decentration [61, 62], anterior band layer fragments [63], and incomplete removal of host Descemet membrane [64], associate with increased graft detachment. Moreover, a study using prospective transplant registry data on 752 DMEK-treated eyes showed with multivariate analysis that surgical difficulty was a risk factor for both rebubbling (Odds Ratio = 2.28, 95% confidence intervals = 1.20–4.33, p = 0.012) and early graft failure (Odds Ratio = 2.93, 95% confidence intervals = 1.42–6.04, p = 0.012) [65]. These observations are notable because graft detachment (or rebubbling, a surrogate measure of graft detachment) associated repeatedly with ECL in three of the five comprehensive studies that have studied this variable [20, 45, 46] (S1 Table). Other studies have also observed this relationship in DMEK [38, 65–68]. However, we did not observe that rebubbling predicted ECL in our study. This may reflect in part the liberal use of rebubbling in our center. Other studies have also failed to detect a relationship between rebubbling and ECL [47, 48, 69].

## Association of patient-related variables with ECL

Patient age was not a predictor of ECL in our study or in five comprehensive studies [43, 44, 46–48] but two comprehensive studies [20, 45], a prospective registry study of 752 eyes [65], and a retrospective study of 66 eyes [70] observed that older patient age associated with more ECL. Dunker et al. proposed that older patients may be more prone to ECL because they find it more difficult to remain lying in the supine position after surgery; this is supported by the fact that graft detachments most often develop inferiorly [65].

We found that patient sex did not associate with ECL. This was also observed by all seven comprehensive studies [20, 43–48].

Our study focused on FECD cases but two comprehensive studies [20, 45] and other studies on cohorts with mixed indications found that severe FECD, or BK rather than FECD, associate with higher ECL [38, 71]. While three other comprehensive studies did not observe this association [43, 44, 48], the aqueous humor of eyes with BK and advanced FECD have high concentrations of proinflammatory cytokines; it has been proposed that this inflammation may damage the endothelial cells in the graft [17].

We observed that patients with low 6-month ECD had significantly lower preoperative ECD, although this was only noted on univariate analysis at 6 months. Nonetheless, two other comprehensive studies have found on multivariate analysis that lower preoperative graft ECD associates with higher ECL [45, 47]. However, this was not observed in a third comprehensive study [48] or a retrospective analysis of 857 eyes [72]. Preoperative ECD may have only a limited impact on ECL if it exceeds a certain threshold [26, 73] (2400 cells/mm$^2$ in our case).

Preoperative visual acuity did not associate with ECL in our study, three comprehensive studies [20, 44, 47], or a prospective study on 108 eyes [68]. It also did not associate with graft detachment in a prospective case series study on 30 eyes [63].

## Association of donor age and sex with ECL

Donor age did not associate with ECL in our study and in four comprehensive studies [43, 45, 46, 48]. This was also observed elsewhere [72]. However, several other studies have suggested that older donors yield wider scrolls that are easier and faster to unroll and may lead to less ECL [59, 74]. Indeed, in our experience, grafts from young donors are generally more difficult to dissect due to high graft elasticity, thus potentially causing difficulties during surgery. It has also been observed that higher preoperative ECDs associate with wider scrolls [59]. Therefore, it is now widely accepted that older donors with high ECD are preferable for DMEK.

We did not study the relationship between donor sex and ECL but the three comprehensive studies that did look at this variable did not observe an association [43, 45, 46].

## Association of surgery duration and triple DMEK/preoperative lens status with ECL

Our 12-month univariate analysis showed that longer surgery duration associated with greater ECL. While there was limited correlation between difficult surgery and surgical duration (r = 0.3, Pearson's correlation test), this finding is compatible with our definition of difficult surgery (any condition hindering the normal course of surgery or requiring more maneuvers than usual). It is also consistent with our finding that difficult surgery associated with longer surgery duration than not-difficult surgery. We did not find other studies on the relationship between ECL and surgery duration. It should be noted that the relatively low correlation between difficult surgery and surgical duration in our study ruled out the possibility that difficult surgery and surgery time were confounding each other in the multivariate analyses.

Triple DMEK (*versus* DMEK-alone) did not associate with ECL in our study, two comprehensive studies [44, 48], and three other studies [71, 75, 76]. However, other studies have found that triple DMEK associates with worse ECL [77] and graft detachment [78]. The reason for this discrepancy is unclear but it may reflect differences between studies in terms of whether DMEK-alone was conducted on phakic or pseudophakic eyes: Siebelmann et al. noted that DMEK on pseudophakic eyes associated with worse rebubbling than triple DMEK and phakic DMEK [79]. Indeed, the comprehensive study of Peraza-Nieves et al. also noted that phakic DMEK associated with better ECL than pseudophakic DMEK [20], although this was not observed in three other comprehensive studies [43–45] or a retrospective analysis of

62 eyes [80]. It has been suggested that pseudophakia may associate with high levels of proinflammatory cytokines in the aqueous humor [81].

## Association between mean anterior keratometry and rebubbling

Interestingly, our side analysis showed that an arched mean anterior keratometry tended to associate with less rebubbling. To our knowledge, the contribution of anterior keratometry to DMEK outcomes has not been studied previously. We speculate that this association may reflect the easier adherence of the DMEK graft (once it is unfolded in the anterior chamber) to a steep cornea rather than to a flat one.

## Study limitations

The present study has several limitations. First, its retrospective design may lend itself to selection and information bias. However, the data were recorded prospectively. Second, it is a monocentric study. However, all surgeries were conducted by one experienced surgeon, which may limit confounding due to surgeon differences. Third, postoperative ECD was measured at the cornea center with specular microscopy: a recent study found that such measurements may not accurately reflect postoperative ECD throughout the entire graft [82]. This could also partly explain the inconsistencies between studies in terms of the relationships between postoperative ECD and pre/perioperative variables. Fourth, since all patients underwent surgery for FECD, our results cannot be extrapolated to DMEK for other endothelial disorders. It will be of interest to determine whether ECL after pseudophakic BK associates with similar pre/perioperative variables. Fifth, it is possible that patient variables that associate with unusual ocular anatomy (*e.g.* high myopia or previous vitrectomy) promote surgical difficulty and ECL. While these variables were rare or not routinely collected in our cohort, they warrant further research.

## Conclusions

Overall, the refractive and endothelial outcomes of our case series were excellent, which supports the notion that DMEK is the surgery of choice for FECD [83] provided that there is no stromal scarring [84]. However, difficulties during surgery may cause lower ECDs at 6 and 12 months. This highlights the importance of meticulous surgical approaches in DMEK and close supervision during the learning curve.

## Supporting information

**S1 Table. Summary of the published studies that comprehensively examined a range of pre/perioperative variables for their ability to predict endothelial cell density/loss after DMEK surgery.**
(DOCX)

## Author Contributions

**Conceptualization:** Jean-Marc Perone.

**Data curation:** Dimitri Chaussard, Florian Bloch, Jean-Charles Vermion, Rémi Moskwa, Jean-Marc Perone.

**Formal analysis:** Dimitri Chaussard, Arpiné Ardzivian Elnar.

**Funding acquisition:** Jean-Marc Perone.

**Investigation:** Dimitri Chaussard, Florian Bloch, Jean-Marc Perone.

**Methodology:** Arpiné Ardzivian Elnar, Jean-Marc Perone.

**Project administration:** Jean-Marc Perone.

**Resources:** Jean-Marc Perone.

**Supervision:** Jean-Marc Perone.

**Validation:** Jean-Marc Perone.

**Visualization:** Dimitri Chaussard, Arpiné Ardzivian Elnar.

**Writing – original draft:** Dimitri Chaussard.

**Writing – review & editing:** Dimitri Chaussard, Florian Bloch, Arpiné Ardzivian Elnar, Yinka Zevering, Jean-Charles Vermion, Rémi Moskwa, Jean-Marc Perone.

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
