## [Decision Letter · Decision Letter 0]

25 Oct 2021

PONE-D-21-31908Factors that predict postoperative endothelial cell density after Descemet membrane endothelial keratoplastyPLOS ONE

Dear Dr. Perone,

Thank you for submitting your manuscript to PLOS ONE. After careful consideration, we feel that it has merit but does not fully meet PLOS ONE’s publication criteria as it currently stands. Therefore, we invite you to submit a revised version of the manuscript that addresses the points raised during the review process.

We look forward to receiving your revised manuscript.

Kind regards,

Hidenaga Kobashi, M.D., Ph.D.

Academic Editor

PLOS ONE

Journal Requirements:

a) Did participants provide their written or verbal informed consent to participate in this study?

Additional Editor Comments:

The reviewers and editor have completed their assessments of your manuscript

Factors that predict postoperative endothelial cell density after Descemet membrane endothelial keratoplasty(PONE-D-21-31908)

and would like to publish it in the journal once you have responded to the referees' comments (enclosed below). In the cover letter with the revised manuscript, please indicate how each of the reviewers' suggestions was addressed.

Reviewers' comments:

Reviewer's Responses to Questions

**Comments to the Author**

1. Is the manuscript technically sound, and do the data support the conclusions?

Reviewer #1: No

Reviewer #2: Yes

2. Has the statistical analysis been performed appropriately and rigorously? 

Reviewer #1: No

Reviewer #2: Yes

3. Have the authors made all data underlying the findings in their manuscript fully available?

Reviewer #1: No

Reviewer #2: Yes

4. Is the manuscript presented in an intelligible fashion and written in standard English?

Reviewer #1: Yes

Reviewer #2: No

5. Review Comments to the Author

Reviewer #1: Authors describe the Factors that predict postoperative endothelial cell density after Descemet membrane endothelial keratoplasty. Although it’s an interesting study, there’re many points that should be addressed.

This may reflect endothelial cell loss due to excessive graft handling and/or an intrinsic unhealthiness of the endothelial cells in the graft that conferred unwanted physical properties onto the graft that complicated its preparation/unfolding.

1) Authors should discuss after dividing into the factors of graft, and those of patients, which means DMEK for eyes after vitrectomy, or with high myopia, or more. The method that authors defined include many confounding factors. At least, authors should be combine or discuss these two factors of difficult graft preparation or unfolding simultaneously. (Line 37, 182)

2) Please let readers know the definition of the surgical time. Do you mean the time from the starting of graft preparation by surgeons, or the patient’s procedure? Does it include the graft preparation time? If so, your surgical time should be diminished by the duration time of graft preparation. Do you use pre-stripped tissues by eyebanks, or non-stripped tissues?

3) Regarding regression analysis, there is no description of R value.

4) The evaluating factors of simple, and multiple regression analysis seem to be same. How did you decide these evaluating factors with evidence? Authors should consult the statistician.

5) Difficult surgery might be the confounding factor of surgery time. Please consider getting rid of confounding factors from them.

6) “We then divided both the 6- and 12-month cohorts into two subgroups by applying a postoperative ECD cut-off of 1000 cells/mm2. This value was selected because it is considered a risk threshold of chronic corneal edema [Line 271-172]” Eyes with low ECD about 1000 have no corneal edema. Please let me know the reason why you decide the ECD of 1000 was the cut-off point clearly.

7) Please consider showing the dataset.

Reviewer #2: In this study, the authors evaluated the risk factors of endothelial cell loss in Fuchs endothelial corneal dystrophy patients post DMEK. The statistical analysis was well performed, and the authors' findings provide important data on the risk factors of endothelial cell loss post DMEK.

Minor Comments:

1. Please enlist the services of a professional English editing service, as numerous sentences in the manuscript are not suitable for use in an academic research paper.

2. Title: The words "Factors that predict …" need to be revised, as the wording is not fully suitable.

3. Line 30: The authors stated "pseudophakic-DMEK or triple-DMEK". Although I understand what the authors wish to state, the wording needs to be revised in order to make it suitable for use in an academic research paper and so that all readers can easily understand.

4. Lines 50-56: This is a 'run-on' sentence that involves multiple improperly connected independent clauses (i.e., complete sentences). Thus, it needs to be revised into 2-3 sentences for improved readability.

5. Lines 104-107: "primary DMEK (if their affected eye was pseudophakic) or triple-DMEK (phacoemulsification followed by DMEK)". Once again, "primary DMEK and triple DMEK" needs to be revised for improved clarity.

6. Line 126: Please indicate the organ culture medium that was used and the storage protocol/status, including the temperature.

7. Table 1: "64//39 (62)". What does "62" indicate?

8. Table 4: "Sex 16 (73%)". Please indicate what "73%" is.

9. Line 315: The statement "difficult surgery" is too vague and needs to be explained in greater detail in order to better clarify the authors' intended meaning.

10. The Discussion section is divided into too many subheadings. For example, Lines 417-420 include only 2 sentences. Thus, please reduce the number of subheadings in the Discussion section for improved readability.

6. PLOS authors have the option to publish the peer review history of their article (what does this mean?). If published, this will include your full peer review and any attached files.

Reviewer #1: No

Reviewer #2: No

---

## [Author Response · Author response to Decision Letter 0]

26 Jan 2022

Point-by-Point Responses to the reviewers

We are very grateful for all reviewers for their thorough review of our paper and their thoughtful comments. We have addressed all points to the best of our ability, as follows. 

Reviewer #1

Authors describe the Factors that predict postoperative endothelial cell density after Descemet membrane endothelial keratoplasty. Although it’s an interesting study, there’re many points that should be addressed.

This may reflect endothelial cell loss due to excessive graft handling and/or an intrinsic unhealthiness of the endothelial cells in the graft that conferred unwanted physical properties onto the graft that complicated its preparation/unfolding.

1) Authors should discuss after dividing into the factors of graft, and those of patients, which means DMEK for eyes after vitrectomy, or with high myopia, or more. 

Reply: To address this comment, we looked at the patients with and without vitrectomy and high myopia, as follows.

Since there were only two eyes that had undergone previous vitrectomy, we did not conduct further analyses on this patient variable. 

We compared the patients with and without high myopia (defined as axial length >25 mm) in terms of any surgical difficulty, difficult graft preparation alone, difficult graft unfolding alone, and ECD at 6 and 12 months. It should be noted that the high myopia case size was small (n=12) and the axial length data were missing in 26 of the 103 cases. As shown in the table below, univariate analysis revealed that the patients with high myopia were significantly more likely to experience difficult graft unfolding (p=0.045). However, high myopia did not associate with greater postoperative ECD loss at 6 months (p=0.56) or 12 months (p=0.84). In the literature, one study has noted that eyes with a *shorter* axial length associate with greater ECL after DMEK (Borroni et al. 2017) but two other studies did not observe an association between axial length and post-DMEK ECL (Inoda et al. 2020; Lekhanont et al. 2021). Given this variability in the literature and the sample size limitations in the univariate analysis below, we decided not to add these data to the paper. Instead, we added the following text to the Study Limitations section: 

“Fifth, it is possible that patient variables that associate with unusual ocular anatomy (e.g. high myopia or previous vitrectomy) promote surgical difficulty and ECL. While these variables were rare or not routinely collected in our cohort, they warrant further research.” Pages 23–24, lines 457–459.

Variables No myopia High myopia Univariate p-value

 Axial length <25 mm Axial length ≥25 mm 

 N = 65 N = 12

Any surgical difficulty 8 (12%) 4 (33%) 0.09

Difficult graft preparation 3 (5%) 1 (8%) 0.50

Difficult graft unfolding 3 (5%) 3 (25%) 0.045

6 postoperative months 1423 ± 406 (690-2500) 1352 ± 503 (700-2180) 0.56

12 postoperative months 1258 ± 397 (509-2400) 1261 ± 505 (688-2180) 0.84

Borroni D, Ferronato M, Krumina Z, Parekh M. Importance of Axial Length and Functional Corneal Endothelial Cells in Descemet Membrane Endothelial Keratoplasty. Cornea. 2017 Dec;36(12):e35-e36. 

Inoda S, Hayashi T, Takahashi H, et al. Factors associated with endothelial cell density loss post Descemet membrane endothelial keratoplasty for bullous keratopathy in Asia. PLoS One. 2020;15(6):e0234202. Published 2020 Jun 11. 

Lekhanont K, Pisitpayat P, Cheewaruangroj N, Jongkhajornpong P, Nonpassopon M, Anothaisintawee T. Outcomes of Descemet Membrane Endothelial Keratoplasty in Bangkok, Thailand. Clin Ophthalmol. 2021 May 31;15:2239-2251. 

1a) The method that authors defined include many confounding factors. At least, authors should be combine or discuss these two factors of difficult graft preparation or unfolding simultaneously. (Line 37, 182)

Reply: The “difficult surgery” variable in Tables 1–5 refers to all 18 cases of difficult surgery, which included seven difficult graft dissection cases, seven difficult graft unfolding cases, and four other cases (two anterior segment hazard cases, one air bubble below the iris case, and one patient movement case). Only in Figure 2 did we examine the three main types of difficult surgery cases separately (graft dissection, graft unfolding, other). 

To address this comment, we altered the Abstract as follows:

“Eighteen eyes involved difficult surgery (14 difficult graft preparation or unfolding cases and four others). Regardless of how the study group was defined, the only pre/perioperative variable that associated significantly with 6- and 12-month ECD was difficult surgery (p=0.01, 0.02, 0.05, and 0.0009). Difficult surgery also associated with longer surgery duration (p=0.002). Difficult-surgery subgroup analysis showed that difficult graft dissection associated with lower postoperative ECD (p=0.03). This association may reflect endothelial cell loss due to excessive graft handling and/or an intrinsic unhealthiness of the endothelial cells in the graft that conferred unwanted physical properties onto the graft that complicated its preparation/unfolding.” Page 3, lines 38–46.

We also changed the Statistics section to make clear that the multivariate analyses were conducted with difficult surgery (not difficult surgery subgroups) and that a separate univariate analysis was conducted with the three difficult surgery subgroups:

“The relationships between 6- and 12-month ECD in the whole cohort and patient, graft, and operative variables (including difficult surgery) were determined with simple and multiple linear regression models. The 6- and 12-month cohorts were also each divided into two subgroups depending on whether their postoperative ECD was <1000 or ≥1000 cells/mm2. This threshold was selected because several ophthalmology societies consider it to be a risk threshold of corneal decompensation [53,54]. The high/low postoperative ECD subgroups were then compared in terms of preoperative/perioperative variables by Mann-Whitney U test (quantitative variables) or Fisher’s exact test (qualitative variables) and then with logistic regression. Subgroup analysis was also conducted to determine whether eyes with different types of surgical difficulties differed from eyes with non-difficult surgery in terms of postoperative ECD: here, Kruskal-Wallis test followed by Mann-Whitney U test was used.” Pages 9–10, lines 189–201.

2) Please let readers know the definition of the surgical time. Do you mean the time from the starting of graft preparation by surgeons, or the patient’s procedure? Does it include the graft preparation time? If so, your surgical time should be diminished by the duration time of graft preparation. Do you use pre-stripped tissues by eyebanks, or non-stripped tissues?

Reply: We dissect the grafts just before the actual surgical procedure. This is indicated by the text in the Methods:

“To prepare the grafts for DMEK, they were first trephined with a Hanna trephine (Busin Punch 17200D 8mm single use; Moria SA, Antony, France) and the Descemet membrane-endothelium complex was dissected off with a disposable curved nontoothed forceps (Single Use Tying Forceps Curved 5mm Platform 17501; Moria SA, Antony, France) under a microscope...” Page 7, lines 130–134

The variable “surgical time” includes the time needed to prepare the graft. Since the term “difficult surgery” includes graft preparation difficulties, we specified both terms in the “Preoperative, perioperative, and postoperative variables” section of the Methods:

“DMEK surgery was labelled as "difficult" if any part of it was not standard. Examples include difficult graft dissection, graft defects, abnormal graft elasticity, difficulties unfolding the scroll in the anterior chamber, or any condition hindering the normal course of surgery or requiring more maneuvers than usual. Surgical time was defined as the time taken from starting graft preparation to placing the suture.” Page 9, lines 182–186.

Please note that the definition of the term “difficult surgery” was moved from the “Graft preparation, surgical techniques, postoperative treatment, and follow-up” Methods section to comply with a request of another reviewer.

3) Regarding regression analysis, there is no description of R value.

Reply: We have provided the R2 values in Tables 2–5 along with other statistics for the simple and multivariate regression analyses.

4) The evaluating factors of simple, and multiple regression analysis seem to be same. How did you decide these evaluating factors with evidence? Authors should consult the statistician.

Reply: All statistics were conducted by a hospital biostatistician, who is also an author (AE). The same 10 factors that were analyzed in the simple regression analyses were also included in the multiple regression analyses because the sample size was sufficient for this number of factors. This point was added to the Statistics section:

“The relationships between 6- and 12-month ECD in the whole cohort and patient, graft, and operative variables (including difficult surgery) were determined with simple and multiple linear regression models. All 10 variables examined in the univariate analyses were considered in the multivariate analyses because the sample size was sufficient for this number of factors.” Pages 9–10, lines 189–193.

5) Difficult surgery might be the confounding factor of surgery time. Please consider getting rid of confounding factors from them.

Reply: The correlation coefficient for difficult surgery and surgery time was 0.3 (Pearson’s test). This suggests that there is not a collinearity problem in the multiple regression analyses. Nonetheless, to check, we repeated the multiple regression analyses (i) with surgical difficulty but not surgical time and (ii) with surgical time but not surgical difficulty. The results were essentially the same:

(i) Surgical difficulty associated/tended to associate with low ECD at six months (whole cohort, p=0.011; high/low preoperative ECD subgroups, p=0.05) and especially at 12 months (p=0.02 and 0.0005) (the equivalent p values in the multiple regression analyses with both difficult surgery and surgical time in Tables 2–5 were 0.010, 0.05, 0.02, and 0.0009, respectively). 

(ii) Associations between surgical time and ECD were not observed at six months (p=0.79 and 0.92) or 12 months (p=0.49 and 0.47) (the equivalent p values in the multiple regression analyses with both difficult surgery and surgical time in Tables 2–5 were 0.54, 0.51, 0.84, and 0.53, respectively). 

To address this point, we added the following text to the Discussion section on surgical time:

“Our 12-month univariate analysis showed that longer surgery duration associated with greater ECL. While there was limited correlation between difficult surgery and surgical duration (r=0.3 , Pearson’s correlation test), this finding is compatible with our definition of difficult surgery (any condition hindering the normal course of surgery or requiring more maneuvers than usual). It is also consistent with our finding that difficult surgery associated with longer surgery duration than not-difficult surgery. We did not find other studies on the relationship between ECL and surgery duration. It should be noted that the relatively low correlation between difficult surgery and surgical duration in our study ruled out the possibility that difficult surgery and surgery time were confounding each other in the multivariate analyses.” Page 22, lines 418–426.

6) “We then divided both the 6- and 12-month cohorts into two subgroups by applying a postoperative ECD cut-off of 1000 cells/mm2. This value was selected because it is considered a risk threshold of chronic corneal edema [Line 271-172]” Eyes with low ECD about 1000 have no corneal edema. Please let me know the reason why you decide the ECD of 1000 was the cut-off point clearly.

Reply: We chose the 1000 cells/mm2 threshold because the American Academy of Ophthalmology indicates that corneas with ECDs below this value are at risk of corneal decompensation [New Ref 53]. The Japanese Corneal Society also considers ECDs of 500–1000 to indicate corneal endothelial damage that could endanger corneal transparency [New Ref 54]. Other studies have therefore utilized this threshold [e.g. Refs A–C below]. To address this, we have: 

(i) Added the following text to the Statistics section of the Methods, removed ref 53, and replaced it with new ref 53 and ref 54:

“This threshold was selected because several ophthalmology societies consider it to be a risk threshold of corneal decompensation [53,54].” Page 10, lines 194–196.

(ii) Altered the original Results text to explain why we used this threshold, removed ref 53, and replaced it with new ref 53 and ref 54:

“We then divided both the 6- and 12-month cohorts into two subgroups by applying a postoperative ECD cut-off of 1000 cells/mm2. This value was selected because several ophthalmology societies consider it to be a risk threshold of corneal decompensation [53,54].” Page 15, 279–280.

New ref 53: Corneal endothelial photography. Three-year revision. American Academy of Ophthalmology. Ophthalmology. 1997 Aug;104(8):1360-5. PMID: 9261327

New ref 54: Kinoshita, Shigeru, et al. "Grading for corneal endothelial damage." Nippon Ganka Gakkai Zasshi (2014): 81-83.

Ref A: Li Y, Fu Z, Liu J, Li M, Zhang Y, Wu X. Corneal Endothelial Characteristics, Central Corneal Thickness, and Intraocular Pressure in a Population of Chinese Age-Related Cataract Patients. J Ophthalmol. 2017;2017:9154626. doi:10.1155/2017/9154626

Ref B: Chen HC, Huang CW, Yeh LK, Hsiao FC, Hsueh YJ, Meir YJ, Chen KJ, Cheng CM, Wu WC. Accelerated Corneal Endothelial Cell Loss after Phacoemulsification in Patients with Mildly Low Endothelial Cell Density. J Clin Med. 2021 May 24;10(11):2270. doi: 10.3390/jcm10112270.

Ref C: Hayashi K, Yoshida M, Manabe S, Hirata A. Cataract surgery in eyes with low corneal endothelial cell density. J Cataract Refract Surg. 2011 Aug;37(8):1419-25. doi: 10.1016/j.jcrs.2011.02.025. Epub 2011 Jun 17. PMID: 21684110.

7) Please consider showing the dataset.

Reply: Unfortunately, the datasets generated during the current study cannot be made publicly available according to French Law No. 2018-493 of June 20, 2018 on the protection of personal data (The General Data Protection Regulation (Regulation (EU) 2016/679) (GDPR: article 9). However, they are available from the corresponding author on reasonable request. This data processing is compliant with a baseline methodology reference methodology (MR-004) for which the our hospital signed a compliance commitment on October 8, 2018.

Reviewer #2

In this study, the authors evaluated the risk factors of endothelial cell loss in Fuchs endothelial corneal dystrophy patients post DMEK. The statistical analysis was well performed, and the authors' findings provide important data on the risk factors of endothelial cell loss post DMEK.

Minor Comments:

1. Please enlist the services of a professional English editing service, as numerous sentences in the manuscript are not suitable for use in an academic research paper.

Reply: One of us (YZ) is a native English speaker with a PhD in the medical field and 25 years of experience in medical and scientific writing and editing. Her website can be viewed on https://www.scimeditor.com/

She has made all changes requested and is grateful to the reviewer for pointing out where the language could be improved or made more precise/readable.

2. Title: The words "Factors that predict …" need to be revised, as the wording is not fully suitable.

Reply: The title has been changed to be more descriptive (underlined texts were added):

“Identification of the preoperative and perioperative factors that predict postoperative endothelial cell density after Deescement membrane endothelial keratoplasty: a retrospective cohort study” Page 1, lines 2–5.

3. Line 30: The authors stated "pseudophakic-DMEK or triple-DMEK". Although I understand what the authors wish to state, the wording needs to be revised in order to make it suitable for use in an academic research paper and so that all readers can easily understand.

Reply: It is important to distinguish between (i) DMEK that immediately follows phacoemulsification (triple-DMEK), (ii) DMEK on eyes that have previously undergone phacoemulsification (what we called pseudophakic-DMEK), and (iii) DMEK on eyes that have not undergone previous phacoemulsification and are not undergoing it during the index surgery (none of our patients fell into this category). Since the latter two surgical regimens are often insufficiently distinguished in the literature, we specified that the patients who did not undergo concomitant phacoemulsification had already previously undergone this procedure. To ensure that this is clear, the terms “triple-DMEK” and “pseudophakic-DMEK” have now been explicitly defined in both the Abstract and Methods section.

Abstract: “This retrospective study was conducted with consecutive adult patients with Fuchs endothelial corneal dystrophy who underwent DMEK in 2015–2019 and were followed for 12 months. Patients either underwent concomitant cataract surgery (triple-DMEK) or had previously undergone cataract surgery (pseudophakic-DMEK).” Page 3, lines 28–32.

Methods: “The study cohort consisted of all consecutive adult (≥18 years) patients with FECD who underwent DMEK between October 2015 and October 2019 and who were followed up for at least 12 months. Patients either underwent concomitant cataract surgery (defined as triple-DMEK) or had previously undergone cataract surgery (defined as pseudophakic-DMEK).” Page 6, lines 106–109.

4. Lines 50-56: This is a 'run-on' sentence that involves multiple improperly connected independent clauses (i.e., complete sentences). Thus, it needs to be revised into 2-3 sentences for improved readability.

Reply: The sentence has been simplified as follows:

“In 1998–2002, endothelial keratoplasty became a feasible alternative to penetrating keratoplasty (PKP) for corneal endothelial disorders such as Fuchs endothelial corneal dystrophy (FECD) and moderate bullous keratopathy (BK). At this timepoint, Melles introduced deep lamellar endothelial keratoplasty (DLEK). This procedure involves (i) dissecting off a posterior lamellar disc from the diseased recipient cornea; (ii) inserting a folded donor disc consisting of posterior stroma, Descemet membrane, and endothelium via a self-healing tunnel incision; and (iii) unfolding the disc and appending it to the recipient cornea with an air bubble [1,2,3]. This method was then further refined by Melles and others...” Page 4, lines 51–58.

6. Line 126: Please indicate the organ culture medium that was used and the storage protocol/status, including the temperature.

Reply: This information (Eurobio medium at 31°C) has been added to the Methods section. Page 7, line 129.

7. Table 1: "64//39 (62)". What does "62" indicate?

Reply: It was intended to indicate “62% of the patients are female”. We have deleted it since it is unnecessary.

8. Table 4: "Sex 16 (73%)". Please indicate what "73%" is.

Reply: It refers to “Female sex” (i.e. 73% of the patients with ECD of <1000 cells/mm2 were female). The word “Female” was added to Tables 4 and 5.

9. Line 315: The statement "difficult surgery" is too vague and needs to be explained in greater detail in order to better clarify the authors' intended meaning.

Reply: The term “difficult surgery” was defined in the Methods as:

“DMEK surgery was labelled as "difficult" if any part of it was not standard. Examples include difficult graft dissection, graft defects, abnormal graft elasticity, difficulties unfolding the scroll in the anterior chamber, or any condition hindering the normal course of surgery or requiring more maneuvers than usual.” Page 9, lines 182–185.

However, this text was originally located in the “Graft Preparation, surgical techniques, postoperative treatment, and follow-up” Methods section. To ensure that the reader can more easily find this definition, this text was moved to the Methods section that is now called “Preoperative, perioperative, and postoperative variables”. 

Moreover, the following text was added to the beginning of the Discussion:

“The present cohort study searched for pre/perioperative factors that shape the postoperative ECD after DMEK. We observed with two statistical methods that difficult surgery associated significantly with lower postoperative ECD, particularly at 12 months. DMEK surgery was deemed difficult if there was any condition that hindered or prolonged the normal surgical course.” Page 18, lines 326–330.

10. The Discussion section is divided into too many subheadings. For example, Lines 417-420 include only 2 sentences. Thus, please reduce the number of subheadings in the Discussion section for improved readability. 

Reply: The subheadings in the Discussion were reduced to six (from nine). Patient-related variables (age, sex, indication, and preoperative ECD and BSCVA) were combined into one section while surgery duration was added to the Triple-DMEK section. Section headings and references were altered to reflect these changes.

---

## [Decision Letter · Decision Letter 1]

10 Feb 2022

Identification of the preoperative and perioperative factors that predict postoperative endothelial cell density after Descemet membrane endothelial keratoplasty: a retrospective cohort study

PONE-D-21-31908R1

Dear Dr. Perone,

We’re pleased to inform you that your manuscript has been judged scientifically suitable for publication and will be formally accepted for publication once it meets all outstanding technical requirements.

Kind regards,

Hidenaga Kobashi, M.D., Ph.D.

Academic Editor

PLOS ONE

Additional Editor Comments (optional):

Congratulations, the reviewers and academic editor agreed the publication in Plos One.

Reviewers' comments:

Reviewer's Responses to Questions

**Comments to the Author**

1. If the authors have adequately addressed your comments raised in a previous round of review and you feel that this manuscript is now acceptable for publication, you may indicate that here to bypass the “Comments to the Author” section, enter your conflict of interest statement in the “Confidential to Editor” section, and submit your "Accept" recommendation.

Reviewer #1: All comments have been addressed

Reviewer #2: All comments have been addressed

2. Is the manuscript technically sound, and do the data support the conclusions?

Reviewer #1: Yes

Reviewer #2: Yes

3. Has the statistical analysis been performed appropriately and rigorously? 

Reviewer #1: Yes

Reviewer #2: Yes

4. Have the authors made all data underlying the findings in their manuscript fully available?

Reviewer #1: Yes

Reviewer #2: Yes

5. Is the manuscript presented in an intelligible fashion and written in standard English?

Reviewer #1: Yes

Reviewer #2: Yes

6. Review Comments to the Author

Reviewer #1: The data presented looks proper.　This paper was well revised.　I would like to expect you to perform further study.

Reviewer #2: I appreciate that authors revised the manuscript based on the comments. I believe that the manuscript was improved and it will be beneficial for the readers in this research field.

7. PLOS authors have the option to publish the peer review history of their article (what does this mean?). If published, this will include your full peer review and any attached files.

Reviewer #1: **Yes: **TAKAHIKO HAYASHI

Reviewer #2: No

---

## [Editor Report · Acceptance letter]

14 Feb 2022

PONE-D-21-31908R1 

Identification of the preoperative and perioperative factors that predict postoperative endothelial cell density after Descemet membrane endothelial keratoplasty: a retrospective cohort study 

Dear Dr. Perone:

I'm pleased to inform you that your manuscript has been deemed suitable for publication in PLOS ONE. Congratulations! Your manuscript is now with our production department. 

Kind regards, 

on behalf of

Dr. Hidenaga Kobashi 

Academic Editor

PLOS ONE